# TANKBind: Trigonometry-Aware Neural NetworKs for Drug-Protein Binding Structure Prediction

**Wei Lu**[*][†]
Galixir Technologies

**Qifeng Wu**[†]
Fudan University

**Jixian Zhang**[†]
Galixir Technologies

**Jiahua Rao**
Sun Yat-sen University

**Chengtao Li**
Galixir Technologies

**Shuangjia Zheng**[*][†]
Galixir Technologies
Sun Yat-sen University

## Abstract

Illuminating interactions between proteins and small drug molecules is a long-standing challenge in the field of drug discovery. Despite the importance of understanding these interactions, most previous works are limited by hand-designed scoring functions and insufficient conformation sampling. The recently-proposed graph neural network-based methods provides alternatives to predict protein-ligand complex conformation in a one-shot manner. However, these methods neglect the geometric constraints of the complex structure and weaken the role of local functional regions. As a result, they might produce unreasonable conformations for challenging targets and generalize poorly to novel proteins. In this paper, we propose Trigonometry-Aware Neural networKs for binding structure prediction, TANKBind, that builds trigonometry constraint as a vigorous inductive bias into the model and explicitly attends to all possible binding sites for each protein by segmenting the whole protein into functional blocks. We construct novel contrastive losses with local region negative sampling to jointly optimize the binding interaction and affinity. Extensive experiments show substantial performance gains in comparison to state-of-the-art physics-based and deep learning-based methods on commonly-used benchmark datasets for both binding structure and affinity predictions with variant settings.

## 1 Introduction

Proteins are the workhorses of human bodies. They have a wide range of interaction partners, small molecules, other proteins, and DNA/RNA, for example. In this paper, we focus on drug-like small molecules as the interaction partners for proteins. The words *ligands*, *drugs*, *small molecules* and *compounds* are used interchangeably throughout the paper. Small molecules activate or inhibit activities of target proteins through mostly non-covalent interactions. In 2021, FDA approved 60 new drugs, among which 36 were small molecules Kinch et al. [2022]. Understanding the mechanism-of-actions and off-target effects of drug molecules typically requires analyzing the structures of the related protein-ligand complexes Boopathi et al. [2021], Xie et al. [2011], but solving the complex structure experimentally is a an extremely challenging task. Despite tremendous effort spent on this topic over the last 50 years, only about 19,000 protein-ligand complex structures have been solved experimentally using X-ray, Cryo-EM or NMR Liu et al. [2015]. On the other hand, the estimated chemical space of drug is $10^{60}$ and estimated number of unique proteins in human body is

---

[*]Correspondance to {wei.lu, shuangjia.zheng}@galixir.com

[†]These authors contribute equally to this work. Qifeng Wu and Jiahua Rao work as interns at Galixir.

36th Conference on Neural Information Processing Systems (NeurIPS 2022).

at least 20, 000, making the number of possible protein-ligand complex far exceeding the number of experimentally solved structures Reymond et al. [2010], Ponomarenko et al. [2016].

On the computational side, molecular docking is a commonly-used method for predicting the protein-ligand complex structures the corresponding binding affinities Trott and Olson [2010], Friesner et al. [2004], Ackloo et al. [2022], Gentile et al. [2022]. Generally, the docking process involves three main stages: (1) locating favorable binding sites given a protein target; (2) sampling the ligand conformation as well as its position and orientation within these sites; (3) scoring and ranking the conformations of the complex using physics-inspired empirical energy functions to refine the structures and assess protein-ligand binding affinity. Due to its good interpretability and usability, docking has been integrated in drug development process for a long time and a number of successful cases have been reported Anderson [2003]. However, most open-source docking packages use atom-level pairwise scoring functions, limiting the capacity to model the many-body effects. Moreover, they need to sample a large range of possible ligand poses and protein side-chain conformations, which leads to relatively high computational cost Trott and Olson [2010], Jain [2006].

To overcome these challenges, we propose a two-stage deep learning framework to neuralize the molecular docking process and predict the binding structures with better accuracy and lower computational cost. In the first stage, we segment the whole protein into functional blocks and predict their interactions with the ligand, creating an protein-ligand interaction energy landscape using a novel trigonometry-aware architecture. The trigonometry module has enough model capacity to capture many-body effects. In the second stage, we prioritize the crystallized binding structures by constrastively ensuring a weaker binding affinity for non-native interactions. In particular, our model improves the drug-protein binding structure predictions with a combination of (i) a novel trigonometry-aware architecture that jointly infuses trigonometry constraints and excluded-volume effects as inductive biases, (ii) a new divide-and-conquer strategy that constructs the protein-ligand local functional binding pairs in a contrastive manner. By doing so, we create a funnel-shape energy landscape for the inter-molecular interaction, removing the need of extensive sampling Jumper et al. [2021], Jain [2006], Chen et al. [2020a], Onuchic et al. [1997].

Our novel method is well-motivated by leveraging prior knowledge from physics and biology. Physically, the inter-molecular trigonometry module, inspired by the intra-molecular Evoformer module used in AlphaFold2 Jumper et al. [2021], ensures that our energy landscape disfavors configurations of protein-ligand complexes that are prohibited by laws of nature, for instance, no two atoms could overlap and the distances between atoms have to satisfy triangle inequality theorem in euclidean geometry. More details on these constraints is shown in section 3.3. Biologically, the functional regions of proteins tend to be more conserved and closely associated with binding De Juan et al. [2013], Glaser et al. [2003], allowing the model to learn critical information and generalize better to unseen proteins.

We evaluate our algorithm against several state-of-the-art deep learning and physics-based docking methods on task of binding structure prediction under multiple settings. Compared with baselines, our model increase the fraction of predictions with ligand root-mean-square deviation (RMSD) less than 5Å by 16% in re-docking setting, 22% in self-docking setting, and 42% in the more difficult new-protein setting. Our model is also capable of predicting binding affinities, achieving better correlations with experimentally-measured values than sequence-based, structure-based and even complex-based methods. We also show that TankBind has the potential to discover novel mechanism-of-actions of drug molecules by identifying unseen protein binding sites.

## 2 Related Work

**Geometric Deep Learning for drug discovery.** There has been a surge of interest in integrating geometric priors for representation learning in the domain of drug discovery Jumper et al. [2021], Baek et al. [2021], Jing et al. [2021], Ganea et al. [2021], Jin et al. [2021], Ingraham et al. [2019], AlQuraishi [2019], Schütt et al. [2017], Somnath et al. [2021]. Recent researches have incorporated geometric information and symmetry properties of the input signals to improve the spatial perception of the learned representations. These works have been shown great potential in various applications like protein structure modeling Jumper et al. [2021], Baek et al. [2021], Jing et al. [2021], Ganea et al. [2021], molecular low-energy generation prediction Shi et al. [2021], Xu et al. [2022], Méndez-Lucio et al. [2021], property/function prediction Schütt et al. [2017], Somnath et al. [2021] and molecule

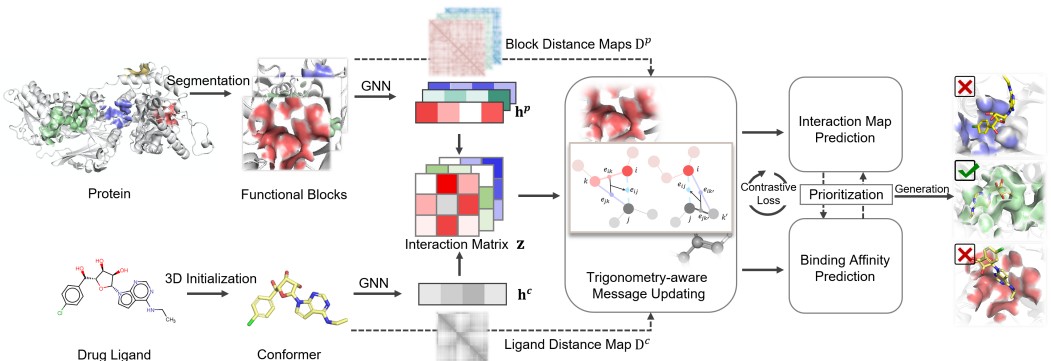

Figure 1: Overview of TankBind Model. The whole protein is divided into blocks of radius 20Å, each block is going through the TankBind model along with the drug compound. Both protein blocks and drug compound are modeled as graphs. The block-compound interaction matrix evolved multiple times with additional input based on the distance maps of the protein block and the compound through trigonometry module. Based on the updated interaction embedding, the model predicts the binding affinity of the compound to the blocks and the block-ligand distance maps. A constrastive loss function is used to ensure the native block binds stronger to the compound than decoys.

design Jin et al. [2021], Ingraham et al. [2019]. Among which, AlphaFold 2 achieved outstanding performance in protein structure prediction Jumper et al. [2021], representing the state-of-the-art geometry-aware method. Our work is inspired from this groundbreaking work, adapting it from the intra-molecular structure prediction to the field of predicting the inter-molecular binding structure and binding affinity.

**Drug-protein Interaction (DPI) prediction.** The goal of DPI prediction is to illustrate the binding structure and binding affinity between protein and ligand. Apart from docking-based approaches Trott and Olson [2010], Friesner et al. [2004], prior machine learning-based works either use complex-free models to predict the binding affinity directly from protein-ligand pairs Wang and Dokholyan [2021], Li et al. [2020], Tsubaki et al. [2019], Gao et al. [2018], Karimi et al. [2019], Zheng et al. [2020] or make predictions through complex structure that has been previously obtained by experimental or docking approaches Jiménez et al. [2018], Lim et al. [2019], Morrone et al. [2020]. The former ones are less interpretable while the latter requires data involved in vast experimental costs and labour. More recently, EquiBind Stärk et al. [2022] takes a new approach by directly predicting the key points on both the protein and the compound, and aligning their key points through the ingeniously designed optimal transport loss. However, this method may generate compound structures clashing with the protein structures and currently lacks the capability to predict the binding affinity, limiting its use in drug discovery. In contrast, our approach has a trigonometry module imposing geometry constraints and a state-of-the-art binding affinity prediction capability.

## 3 TankBind Model

### 3.1 Overview of TankBind model

The general protocol of our model is shown in figure 1. The encoding of protein and compound is described in section 3.2. The rationale and implementation of trigonometry module is detailed in section 3.3. The design of loss functions for training is described in section 3.4. The generation of atom coordinates from predicted inter-molecular distance map is introduced in section 3.5.

### 3.2 Structural encoders of protein and drug

Our model input is the separate structures of a protein and a drug compound, both encoded as graphs. Indices $i, k$ always operate on the residue dimension, $j, k'$ always on the compound dimension. $n$ is the number of protein nodes and $m$ is the number of compound nodes.

**Protein.** The protein is represented as a proximity 3D graph following Jing et al. [2020]. We denote the protein graph as $\mathcal{G}^p = (\mathcal{V}^p, \mathcal{E}^p)$, where each node $\mathfrak{v}_i^p \in \mathcal{V}^p$ corresponds to an amino acid, and has feature $\mathbf{h}_{\mathfrak{v}^p}^{(i)}$ with both scalar and vector features. Each node also has a position $\mathbf{x}_i^p \in \mathbb{R}^3$ equal to the Cartesian coordinate of $C_{\alpha_i}$. An edge $\mathfrak{e}_{ik}^p$ exists if $\mathfrak{v}_i^p$ is among the 30 nearest neighbors of $\mathfrak{v}_k^p$. Each edge $\mathfrak{e}_{ik}^p \in \mathcal{E}^p$ also encodes both the scalar and the vector features. We then apply the geometric vector perceptrons (GVP) Jing et al. [2020, 2021] to embed the protein and arrive at feature $\mathbf{h}^p \in \mathbb{R}^{n \times s}$ after graph propagation, where $n$ is the number of nodes and $s$ is the embedding size.

To implicitly model side-chain flexibility, we choose a residue-level representation ignoring the finer details of protein structure, separating our method from other methods that use all-atoms or surface vertexes representation Jiang et al. [2021], Gainza et al. [2020]. Also, as shown by Jumper et al. [2021, 2018], residue-level embedding is enough to infer the side-chain conformation.

Motivated by protein co-evolution De Juan et al. [2013] and divide-and-conquer theory, the protein graph, $\mathcal{G}^p$, is further divided into subgraphs $\mathcal{G}^{p'}$. Each subgraph $\mathcal{G}^{p'}$ includes all the $\mathfrak{v}_i^p$ and $\mathfrak{e}_{ij}^p$ inside the functional block. The subgraph is denoted as $\mathcal{G}^{p'} = (\{\mathfrak{v}_i^p, \mathfrak{e}_{ik}^p\} \mid \|\mathbf{x}_i^p - \mathbf{x}_o\| \leq 20\text{Å}, \|\mathbf{x}_k^p - \mathbf{x}_o\| \leq 20\text{Å})$, where $\mathbf{x}_o$ is the center of the functional block predicted by a widely-used ligand-agnostic method, P2rank (published in 2018)Krivák and Hoksza [2018]. Justification for the size of radius and use of P2rank is described in appendix G.

**Drug compound.** The drug compound is represented as a graph using TorchDrug toolkit Zhu et al. [2022]. The compound graph is denoted as $\mathcal{G}^c = (\mathcal{V}^c, \mathcal{E}^c)$ where each node $\mathfrak{v}_j^c \in \mathcal{V}^c$ corresponds to a heavy atom (non-hydrogen atom), and has feature $\mathbf{h}_{\mathfrak{v}^c}^{(j)}$ and each edge $\mathfrak{e}_{jk'}^c$ has feature $\mathbf{h}_{\mathfrak{e}^c}^{(jk')}$. We use Graph Isomorphism Network (GIN) Xu et al. [2018] to embed the compound and arrive at feature $\mathbf{h}^c \in \mathbb{R}^{m \times s}$ after graph propagation, where $m$ is the number of heavy atoms and $s$ is the embedding size.

### 3.3 Details of trigonometry module

The compound feature, $\mathbf{h}^c$, and the protein block feature, $\mathbf{h}^p$, are used to form the initial interaction embedding $\mathbf{z}^{(0)} \in \mathbb{R}^{n \times m \times s}$, $\mathbf{z}_{ij}^{(0)} = \mathbf{h}_i^p \odot \mathbf{h}_j^c$. The interaction embedding will be further updated with pair distance map of protein nodes, $D_{ik}^p = \|\mathbf{x}_i^p - \mathbf{x}_k^p\|$ and pair distance map of compound nodes, $D_{jk'}^c = \|\mathbf{x}_j^c - \mathbf{x}_{k'}^c\|$.

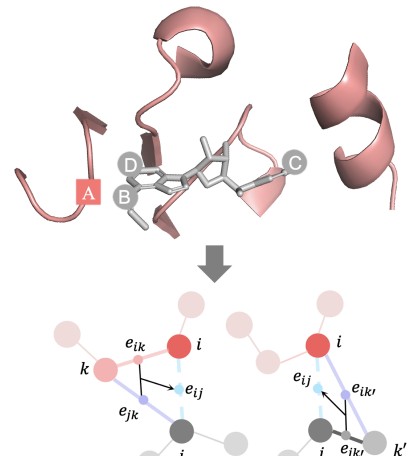

The rationale for including both the pair distance map of the protein nodes and the pair distance map of the compound nodes in updating the protein-compound interaction embedding is explained with two simplified examples. As shown in the upper part of figure 2, if a protein node A is in close proximity with compound node B, then compound node C will not be in contact with node A due to the large distance constraint between node B and C. Distance constraint between compound nodes B and D could also force a node D to be in close contact with protein node A.

Figure 2: Rationale for including trigonometry module. Upper: Protein node in square, compound nodes in circles. Lower: Trigonometry module ensures that the interaction between protein node $i$ and compound node $j$ depends on all protein and compound nodes $k, k'$.

To build this observation of trigonometry constraints into our model, we design the following module to update the interaction embedding, in layer $\ell$, $\forall(i, j)$:

$$\tilde{\mathbf{z}}_{ij}^{(\ell)} = \mathbf{z}_{ij}^{(\ell)} + \Phi\left(\sum_{k=1}^{n} \mathbf{p}_{ik} \mathbf{t}_{kj}^{(\ell)} + \sum_{k'=1}^{m} \mathbf{t}'^{(\ell)}_{ik'} \mathbf{c}_{k'j}\right) \odot \mathbf{g}(\mathbf{z}_{ij}^{(\ell)}) \tag{1}$$

where $\mathbf{p}_{ik} = \phi(D_{ik}^p)$ is the linear embedding of encoded pair distance between protein nodes. $\mathbf{p} \in \mathbb{R}^{n \times n \times s}$, $n$ is the number of nodes in protein block, $s$ is the embedding size. $\mathbf{c}_{jk'} = \phi(D_{jk'}^c)$ is the linear embedding of encoded pair distance between compound nodes. $\mathbf{c} \in \mathbb{R}^{m \times m \times s}$, $m$ is the number of compound nodes. $\mathbf{t}_{ij}^{(\ell)}$ and $\mathbf{t}'^{(\ell)}_{ij}$ are the same gated linear transformations of $\mathbf{z}_{ij}^{(\ell)}$ but with non-

shared parameters, $\mathbf{t}_{ij}^{(\ell)} = \text{Linear}(\mathbf{z}_{ij}^{(\ell)}) \odot \mathbf{g}(\mathbf{z}_{ij}^{(\ell)})$, $\mathbf{t}^{(\ell)} \in \mathbb{R}^{n \times m \times s}$, $\mathbf{g}(\mathbf{z}_{ij}^{(\ell)}) = \text{sigmoid}(\text{Linear}(\mathbf{z}_{ij}^{(\ell)}))$, $\Phi$ is a layernorm function followed by a linear transformation.

Another type of physical constraint need to be take into consideration is the excluded-volume (Van Der Waals) and saturation effect. As shown in the upper figure 2, if protein node A forms a strong interaction, hydrogen bonding for example, with compound node B, then node D is unlikely to form the same type of interaction with node A because node A has limited number of hydrogen donors or acceptors. To account for these effects, we designed a self-attention module to modulate the interaction between a protein node and all compound nodes by taking the whole interaction between this protein node and all compound nodes into consideration.

$$\dot{\mathbf{z}}_{ij}^{(\ell)} = \tilde{\mathbf{z}}_{ij}^{(\ell)} + \Phi(\text{concat}_h(\sum_{k'=1}^{m}(w_{ijk'}^{(\ell)h}\mathbf{v}_{ik'}^{(\ell)h}) \odot \mathbf{g}^h(\tilde{\mathbf{z}}_{ij}^{(\ell)}))) \tag{2}$$

$$w_{ijk'}^{(\ell)h} = \text{softmax}_{k'}(\mathbf{q}_{ij}^{(\ell)h^\top}\mathbf{k}_{ik'}^{(\ell)h}) \tag{3}$$

, where $\mathbf{q}_{ij}^{(\ell)h}, \mathbf{k}_{ij}^{(\ell)h}, \mathbf{v}_{ij}^{(\ell)h}$ are linear transformation of $\tilde{\mathbf{z}}_{ij}^{(\ell)}$, $h$ is number of attention heads. Function $\mathbf{g}^h$ is the standard $\mathbf{g}$ with reshaping the embedding into heads at the end, $\Phi$ is a linear transformation.

Lastly, a non-linear transition module is added to transit the interaction embedding to the next layer through multilayer perceptron, $\mathbf{z}_{ij}^{(\ell+1)} = \text{MLP}(\dot{\mathbf{z}}_{ij}^{(\ell)})$. The whole trigonometry module is composed of three consecutive parts, the trigonometry update, the self-attention modulation, and the non-linear transition module. Layernorm is applied on every input $\mathbf{z}_{ij}^{(\ell)}$ and a 25% dropout is applied to the trigonometry update and self-attention modulation during training. The final outputs, drug-protein binding affinity, $\hat{a} = \sum_{i=1}^{n}\sum_{j=1}^{m}\text{Linear}(\mathbf{z}_{ij}^{(L)})$ , and inter-molecular distance map, $D_{ij}^{pred} = \mathbf{g}(\mathbf{z}_{ij}^{(L)})\text{Linear}(\mathbf{z}_{ij}^{(L)})$, are predicted directly based on the last layer embedding $\mathbf{z}_{ij}^{(L)}$, where $L$ is the number of module stacks.

### 3.4 Design of binding interaction and affinity loss functions

Many previous works model the interaction between compound and protein by only preserving the interaction region, residues that far away are ignored Townshend et al. [2020], Méndez-Lucio et al. [2021]. On the positive side, the computation and memory demand for characterize the interaction between protein and the drug compound is greatly reduced by focusing on regional interaction. But the fact of not binding to alternative binding sites is also a valuable information. By the nature of crystallization, if a protein-compound complex could be successfully crystallized, other possible binding sites on this protein definitely bind less strongly than the native binding site to the compound, therefore, those other binding sites could be used as high-valued decoys. Based on this observation, we designed a max-margin constrastive affinity loss, equation 4, following the idea of Hadsell et al. [2006]. Such that the compound's predicted affinity, $\hat{a}$, to the decoys is less than the experimentally measured affinity, $a$, by a margin value, $\epsilon$.

$$\mathcal{L}_{\text{affinity}}(\hat{a}_\zeta, a) = \mathbb{1}(\zeta)(\hat{a}_\zeta - a)^2 + (1 - \mathbb{1}(\zeta))\max(0, \hat{a}_\zeta - (a - \epsilon))^2 \tag{4}$$

where $\hat{a}_\zeta$ is the predicted affinity to block $\zeta$, and indicator function $\mathbb{1}(\zeta) = 1$ when block $\zeta$ encloses the native ligand, and $\mathbb{1}(\zeta) = 0$ otherwise. We, therefore, take full use of information stored in the whole protein instead of only the native binding region. We also include a mean squared erorr (MSE) loss for native interaction distance map, $\mathcal{L}_{\text{distance}} = \mathbb{1}(\zeta)\frac{1}{nm}\sum_{i=1}^{n}\sum_{j=1}^{m}(D_{ij}^{pred} - D_{ij})^2$. The overall training objective of TankBind is: $\mathcal{L} = \mathcal{L}_{\text{affinity}} + \mathcal{L}_{\text{distance}}$.

### 3.5 Generation of drug coordinates based on predicted inter-molecular distance map.

The Cartesian coordinates, $\{\hat{\mathbf{x}}_j^c\}$, of the heavy atoms of a drug compound could be deduced analytically based on the predicted inter-molecular distance matrix, $D_{ij}^{pred}$, the coordinates of protein nodes, $\{\mathbf{x}_i^p\}$, and the pair distance matrix of compound nodes, $D_{jk'}^c$ Masters et al. [2022], Hoffmann and Noé [2019]. But since predicted distance matrix contains noise, we take a numerical approach Masters et al. [2022], Zsoldos et al. [2007]. By minimizing the total loss, $\mathcal{L}_{\text{generation}}$, which consists of two parts, the interaction loss and the compound configuration loss, we could derive the coordinates

Table 1: Blind self-docking. All models take a pair of ligand structure (generated by RDKit) and protein structure as input, trying to predict the atom coordinates of the ligand after binding. In blind docking, the protein binding site is assumed unknown. Test set is composed of 363 protein-ligand structure crystallized after 2019 curated by PDBbind database. Details about model runtime and the number of model parameters are in appendix C

| | Ligand RMSD | | | | | | Centroid Distance | | | | | |
| | Percentiles ↓ | | | | % Below Threshold ↑ | | Percentiles ↓ | | | | % Below Threshold ↑ | |
| **Methods** | 25% | 50% | 75% | Mean | 2Å | 5Å | 25% | 50% | 75% | Mean | 2Å | 5Å |
|---|---|---|---|---|---|---|---|---|---|---|---|---|
| QVINA-W | 2.5 | 7.7 | 23.7 | 13.6 | 20.9 | 40.2 | 0.9 | 3.7 | 22.9 | 11.9 | 41.0 | 54.6 |
| GNINA | 2.8 | 8.7 | 22.1 | 13.3 | 21.2 | 37.1 | 1.0 | 4.5 | 21.2 | 11.5 | 36.0 | 52.0 |
| SMINA | 3.8 | 8.1 | 17.9 | 12.1 | 13.5 | 33.9 | 1.3 | 3.7 | 16.2 | 9.8 | 38.0 | 55.9 |
| GLIDE(c.) | 2.6 | 9.3 | 28.1 | 16.2 | **21.8** | 33.6 | **0.8** | 5.6 | 26.9 | 14.4 | 36.1 | 48.7 |
| VINA | 5.7 | 10.7 | 21.4 | 14.7 | 5.5 | 21.2 | 1.9 | 6.2 | 20.1 | 12.1 | 26.5 | 47.1 |
| EQUIBIND-U | 3.3 | 5.7 | 9.7 | 7.8 | 7.2 | 42.4 | 1.3 | 2.6 | 7.4 | 5.6 | 40.0 | 67.5 |
| EQUIBIND | 3.8 | 6.2 | 10.3 | 8.2 | 5.5 | 39.1 | 1.3 | 2.6 | 7.4 | 5.6 | 40.0 | 67.5 |
| TANKBind-R | 2.8 | 5.2 | 11.2 | 9.4 | 16.0 | 47.9 | 1.0 | 2.3 | 7.7 | 7.3 | 44.9 | 69.4 |
| TANKBind-C | 2.4 | 4.5 | 8.4 | 8.2 | 19.6 | 54.8 | 0.9 | 1.9 | 5.4 | 6.3 | 53.2 | 73.3 |
| TANKBind-P | 2.6 | 4.5 | 8.1 | 8.5 | 16.3 | 54.0 | 0.9 | 1.9 | 5.2 | 6.4 | 53.2 | 74.4 |
| TANKBind | **2.4** | **4.0** | **7.7** | **7.4** | 19.3 | **61.7** | 0.9 | **1.7** | **4.2** | **5.5** | **56.5** | **77.4** |

of the docked drug coordinates, $\{\hat{\mathbf{x}}_j^c\}$.

$$\mathcal{L}_{\text{generation}} = \mathcal{L}_{\text{interaction}} + \mathcal{L}_{\text{configuration}} = \sum_i^n \sum_j^m (|\hat{D}_{ij} - D_{ij}^{pred}|) + \sum_j^m \sum_{k'}^m (|\hat{D}_{jk'}^c - D_{jk'}^c|) \quad (5)$$

$$\hat{D}_{ij} = \left\| \mathbf{x}_i^p - \hat{\mathbf{x}}_j^c \right\|, \hat{D}_{jk'}^c = \left\| \hat{\mathbf{x}}_j^c - \hat{\mathbf{x}}_{k'}^c \right\| \quad (6)$$

where $n$ is the number of protein nodes, and $m$ is number of compound nodes, and $\{\mathbf{x}_j^p\}$ are the Cartesian coordinates of protein nodes. All inter-molecular distances are clamped to have an upper bound of 10Å to focus on the direct interaction. In self-docking setting, when the compound configuration is unknown, we add a local atomic structures (LAS) mask to the configuration loss to allow for compound flexibility while enforcing basic geometric constraint, $\mathcal{L}_{\text{configuration}} = \sum_j^m \sum_{k'}^m \mathbb{1}(j, k')(|\hat{D}_{jk'}^c - D_{jk'}^c|)$ where $\mathbb{1}(j, k') = 1$ when compound atom $j$ and $k'$ are connected by connected by a bond, or 2-hop away, or in the same ring structure, and $\mathbb{1}(j, k') = 0$ otherwise Stärk et al. [2022], Trott and Olson [2010]. For every test protein-ligand pair, TankBind predicts the binding affinity of the ligand to all segmented functional blocks and chooses the one with strongest affinity to generate the binding structures.

## 4 Evaluation

### 4.1 Protein-ligand binding structure prediction

**Dataset.** We used publicly available PDBbind v2020 dataset Liu et al. [2015] which has the structures of 19443 protein-ligand complexes along with their experimentally measured binding affinity. PDBbind is a database curated based on the Protein Data Bank (PDB) Burley et al. [2021]. We followed the same time split as defined in EquiBind paper Stärk et al. [2022] in which the training and validation data are the protein-ligand complex structures deposited before 2019 and the test set is the structures deposited after 2019. After removing a few structures that unable to process using RDKit from the training set, we had 17787 structures for training, 968 for validation and 363 for testing Landrum et al. [2013]. We also reduced the possibility of encountering equally valid binding sites by removing chains that have no atom within 10Å from any atom of the ligand following the protocol described in Stärk et al. [2022].

**Baselines.** We compared TankBind with the most widely-used docking method AutoDock Vina Trott and Olson [2010] and the recent proposed geometry-based DL method EquiBind Stärk et al. [2022]. We also included four popular docking methods QVina-W, GINA McNutt et al. [2021], SMINA Koes et al. [2013] and GLIDE Friesner et al. [2004] as listed in Stärk et al. [2022].

Table 2: Blind self-docking for unseen receptors. All models evaluated on 142 crystallized protein-compound structures where the proteins have not been observed in training set.

| | Ligand RMSD | | | | | | Centroid Distance | | | | | |
| | Percentiles ↓ | | | | % Below Threshold ↑ | | Percentiles ↓ | | | | % Below Threshold ↑ | |
| Methods | 25% | 50% | 75% | Mean | 2Å | 5Å | 25% | 50% | 75% | Mean | 2Å | 5Å |
|---|---|---|---|---|---|---|---|---|---|---|---|---|
| QVINA-W | 3.4 | 10.3 | 28.1 | 16.9 | 15.3 | 31.9 | 1.3 | 6.5 | 26.8 | 15.2 | 35.4 | 47.9 |
| GNINA | 4.5 | 13.4 | 27.8 | 16.7 | 13.9 | 27.8 | 2.0 | 10.1 | 27.0 | 15.1 | 25.7 | 39.5 |
| SMINA | 4.8 | 10.9 | 26.0 | 15.7 | 9.0 | 25.7 | 1.6 | 6.5 | 25.7 | 13.6 | 29.9 | 41.7 |
| GLIDE | 3.4 | 18.0 | 31.4 | 19.6 | **19.6** | 28.7 | **1.1** | 17.6 | 29.1 | 18.1 | 29.4 | 40.6 |
| VINA | 7.9 | 16.6 | 27.1 | 18.7 | 1.4 | 12.0 | 2.4 | 15.7 | 26.2 | 16.1 | 20.4 | 37.3 |
| EQUIBIND-U | 5.7 | 8.8 | 14.1 | 11.0 | 1.4 | 21.5 | 2.6 | 6.3 | 12.9 | 8.9 | 16.7 | 43.8 |
| EQUIBIND | 5.9 | 9.1 | 14.3 | 11.3 | 0.7 | 18.8 | 2.6 | 6.3 | 12.9 | 8.9 | 16.7 | 43.8 |
| TANKBind-R | 3.6 | 6.9 | 17.0 | 12.6 | 5.6 | 35.2 | 1.3 | 3.6 | 15.7 | 10.3 | 35.2 | 58.5 |
| TANKBind-C | 3.4 | 5.5 | 9.8 | 9.9 | 9.2 | 43.0 | **1.1** | 2.6 | 8.1 | 7.9 | 46.5 | 65.5 |
| TANKBind-P | 3.3 | 5.5 | 10.9 | 11.2 | 5.6 | 45.1 | 1.3 | 2.3 | 7.9 | 9.1 | **47.9** | 66.9 |
| TANKBind | **2.9** | **4.7** | **8.8** | **9.1** | 4.9 | **55.6** | 1.3 | **2.3** | **4.8** | **7.0** | 45.1 | **75.4** |

**Evaluation metrics.** We follow prior work Stärk et al. [2022] and use ligand root-mean-square deviation (RMSD) of atomic positions and centroid distance to compare predicted binding structures with ground-truths. The Ligand RMSD calculates the normalized Frobenius norm of the two corresponding matrices of ligand coordinates. The centroid distance is defined as the the distance between the averaged 3D coordinates of the predicted and ground-truth bound ligand atoms, indicating the model capability of identifying correct binding region. Hydrogens are not involved in the calculation.

**Performance in blind flexible self-docking** We start with a real-world blind self-docking experiment, in which the ligand conformation is not fixed, and the result of re-docking experiment, in which the native ligand conformation is given, is reported in Appendix A. As shown in the table 1, TankBind achieves state-of-the-art performance, outperforming geometry DL-based model EquiBind. This advantage is particularly evident in the top 25% and top 50% ligand RMSD, which allows our method to predict 22% more qualified (below Threshold 5Å) binding poses than EquiBind. This results are also consistent in the metrics of centroid distance, demonstrating that our method also has a clear advantage in the identification of binding region. Even though GLIDE (commercial) and Autodock Vina are established docking software with more than a decade of continuous development, our model remarkably frequently outperforms them. At the same time, we are orders of magnitude faster than them, and on the same level as EquiBind (Appendix C). In addition, we explore the possible of TankBind-R, where we randomly segment the protein, TankBind-P, where we only doing the summation over protein nodes in equation 1, and TankBind-C, where we only sum over compound nodes. The performance reduction on the these variants supports our view that trigonometry message passing between proteins and ligand and segmentation choice are critical to the prediction of binding structures.

**Performance in self-docking unseen protein** We next focus on the new protein setting, in which the tested proteins have not been observed in the training set. Table 2 shows that Tankbind leads to larger improvements over EquiBind and docking methods with regard to ligand-RMSD and centriod distance. This is in line with our expectation that TankBind has better generalization ability due to the physical-inspired trigonometry module and explicit consideration of conservative functional blocks. In this setting, as shown in Figure 3 and table 2, for fractions smaller than 2Å, 5Å and 15Å, the performance between EquiBind and other docking method are comparable, while TankBind is always better by a large margin, further confirming the effectiveness of our method and indicating that the proposed strategy has practical values for the virtual screening of new proteins.

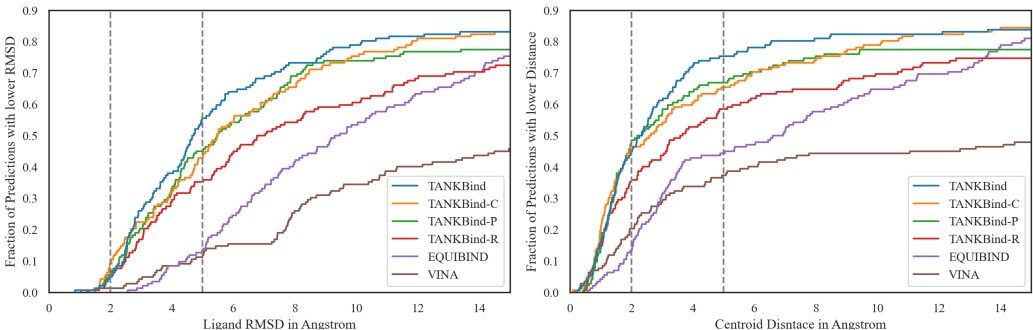

Figure 3: Estimator of the Cumulative Distribution Function (ECDF) plot for ligand RMSD (left) and Centroid Distance (right) from result evaluated on new receptors subset. The x axis of the figure stops at 15Å because comparison for larger RMSD is less meaningful when the predicted location of the ligand is away from the true binding site, a RMSD of 15Å is not better than RMSD of 50Å.

| Methods | RMSE↓ | Pearson↑ | Spearman↑ | MAE↓ |
|---|---|---|---|---|
| TransCPI | 1.741 | 0.576 | 0.540 | 1.404 |
| MONN | 1.438 | 0.624 | 0.589 | 1.143 |
| PIGNet※ | 2.640 | 0.511 | 0.489 | 2.110 |
| IGN | 1.433 | 0.698 | 0.641 | 1.169 |
| HOLOPROT | 1.546 | 0.602 | 0.571 | 1.208 |
| STAMPDPI | 1.658 | 0.545 | 0.411 | 1.325 |
| TANKBind | **1.346** | **0.726** | **0.703** | **1.070** |

Table 3: Binding affinity prediction. TankBind achieves SOTA on all four metrics.

| Methods | Ligand↓ | Centroid↓ | Below2A↑ | Below5A↑ |
|---|---|---|---|---|
| w/o P2Rank | 9.37 | 7.30 | 44.90 | 69.42 |
| w/o Trig | 8.73 | 6.44 | 44.08 | 74.93 |
| TAPE | 8.81 | 6.89 | 50.69 | 73.00 |
| GAT | 8.27 | 6.23 | 56.47 | **78.51** |
| TankBind-P | 8.47 | 6.44 | 53.17 | 74.38 |
| TankBind-C | 8.20 | 6.27 | 53.17 | 73.28 |
| Origin | **7.43** | **5.51** | **56.47** | 77.41 |

Table 4: Ablation results. We listed four main metrics here, a complete table is in appendix E

## 4.2 Protein-ligand binding affinity prediction

TankBind is also capable of predicting protein-ligand binding affinity because of the constrastive affinity loss function. Since we segmented the whole protein into protein blocks, the predicted binding affinity of ligand to the whole protein is equal to the binding affinity to the one protein block that predicted to bind strongest with the ligand. To demonstrate the ability, we compared TankBind with the state-of-the-art binding affinity prediction models.

**Dataset.** We split the dataset into training, test and validation splits based on the same time split described earlier. The experimentally measured affinity data in PDBbind dataset has three different names, depending on the exact experiment setups, 50% inhibiting concentration (IC50), inhibition constant ($K_i$), and dissociation constant ($K_D$), all converted to the unit of molar concentration. Similar to previous methods Somnath et al. [2021], Townshend et al. [2020], we predict negative log-transformed binding affinity.

**Baselines and evaluation metrics.** We compare TankBind against two state-of-the-art sequence-based methods, TransformerCPI Chen et al. [2020b] and MONN Li et al. [2020], two complex-based methods, IGN Jiang et al. [2021] and PIGNet Moon et al. [2022] both requiring prior knowledge of the inter-molecular structure to predict affinity, and two structure-based methods, HOLOPTOT Somnath et al. [2021] and STAMPDPI Wang et al. [2022]. For evaluating various methods, we use four metrics – root mean squared error (RMSE), Pearson correlation coefficient, Spearman correlation coefficient and mean absolute error (MAE). We also include the mean and standard deviation across 3 experimental runs in appendix D.

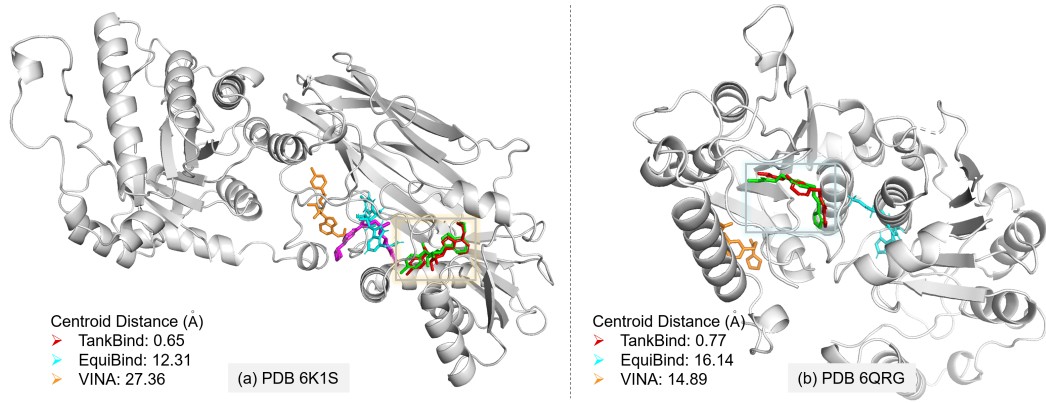

Figure 4: (a) An example of TankBind identifying an unseen binding site. The protein is shown in white, co-crystallized compounds of three PDBs in the training set is shown in purple. The ligand of 6K1S is shown in green. TankBind is able to find this correct pose for the compound, shown in red, while the other two, Vina in orange, and Equibind in cyan, place the compound away from the true binding site. (b) For PDB 6QRG, both protein and compound have not been seen in the training set. But TankBind still find the correct pose. Crystallized ligand colored in green, TankBind prediction in red, EquiBind in cyan and Vina result in organ.

**Result** As shown in Table 3, our model obtains the best performance in PDBbind test set, consistently outperforms SOTA binding affinity prediction methods. Note that even without the prior interaction information, TankBind also achieves better result than complex-based methods (PIGNET and IGN), proving that the predicted binding structural information provided considerable gain to the affinity prediction task.

### 4.3 Ablation study

We conducted ablation studies to investigate factors that influence the performance of proposed TankBind framework. As shown in Table 4, the original version of TankBind with the trigonometry message passing between protein and ligand shows the best performance among all architectures. Replacing the P2rank with a randomly split of blocks performed the worst, which verifies our hypothesis that functional block segmentation can improve generalization. Simple architecture substitutions for protein (TAPE) Rao et al. [2019] and molecular representation (GAT) Veličković et al. [2017] decrease slightly the model performance. Replacing the intra-trigonometry module with the uni-modal variants (TankBind-P and TankBind-C) both caused noticeable decreases in performances.

### 4.4 Case studies

**TankBind correctly identifies an unseen binding site for a new drug compound.** As a representative case, in PDB 6K1S, a seen protein binds to a new drug compound at a site that has not been observed before. This protein has three co-crystallized complex structures in the training set, PDB 4X60, 4X61, 4X63. As shown in the left of figure 4, our method, shown in red, aligns well with the true ligand, shown in green, despite our method has never seen any compound locates at this site before. While other two methods, EquiBind in cyan, Vina in orange identify an incorrect site for this compound. Packages Kalign, Biopython, and Smith-Waterman library are used to systematically analyze the results Lassmann [2020], Cock et al. [2009], Li et al. [2020], Zhao et al. [2013] (see Appendix H).

**TankBind finds the correct pose when both compound and protein are unseen.** We picked two representative examples with both compound and protein are unseen, one, PDB 6QRG, in the right of figure 4 and another, PDB 6KQI, in appendix B. Both PDB 6QRG and PDB 6KQI have max protein

similarity below 0.8 (6QRG 0.78, 6KQI 0.57), and max compound similarity below 0.4 (6QRG 0.36, 6KQI 0.27).

## 5 Conclusion

In this work, we propose a novel binding structure and affinity prediction model, TankBind, that builds trigonometry constraints into the model and explicitly attends to all possible binding sites by segmenting the whole protein into functional blocks. We observe significant improvements on task of binding structure prediction over existing deep learning methods: a 22% increase in the fraction of prediction below 5Å in ligand RMSD, and a 42% increase when the proteins have not been observed in the training set. Moreover, we demonstrate that the model is able to predict affinity and outperform SOTA methods on PDBbind. This work opens a new direction for modelling the inter-molecular interaction between protein and drug molecule. Numerous directions for further exploration include incorporating a ligand conformer generation module, enhancing the dataset with AlphaFold-predicted structure and public available SAR data, integrating the segmentation of functional block in an end-to-end manner, and combining the model with protein backbone dynamics modeling to handle larger scale conformation changes induced by drug-protein interactions.

## Acknowledgments and Disclosure of Funding

We thank Prof.Yuedong Yang, Dr.Leilei Shi, Dr. Jiahui Tong for their helpful discussions; Penglei Wang for his support in binding affinity experiments; Meihui Song for her support in figure drawing. We thank the Guangzhou National Supercomputer Center for providing computational source. S. Z acknowledges support from the National Key R&D Program of China (2020YFB0204803) and National Natural Science Foundation of China (21773313 and 61772566).

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
