# A   Additional binding structure prediction results and pseudo code for trigonometry module

## A.1   Blind re-docking.

In re-docking, the ligand conformations are given as input. As shown in table 5, In 64.5 % of the test set, TankBind has ligand RMSD less than 5Å, compared to 48.2 % for EquiBind and 26.7 % for Vina.

Table 5: Blind re-docking. In re-docking setting, the input compound conformations are the co-crystallized conformations. All three models evaluated on the test set of 363 protein-compound structures crystallized after 2019 curated by PDBbind database.

| | **Ligand RMSD** | | | | | | **Centroid Distance** | | | | | |
| | Percentiles ↓ | | | | % Below Threshold ↑ | | Percentiles ↓ | | | | % Below Threshold ↑ | |
| **Methods** | 25% | 50% | 75% | Mean | 2Å | 5Å | 25% | 50% | 75% | Mean | 2Å | 5Å |
|---|---|---|---|---|---|---|---|---|---|---|---|---|
| QVINA-W | 1.6 | 7.9 | 24.1 | 13.4 | 27.7 | 39.0 | 0.9 | 3.8 | 23.2 | 11.8 | 40.4 | 55.4 |
| GININA | 1.3 | 6.1 | 22.9 | 12.2 | 32.2 | 46.8 | 0.7 | 2.8 | 22.1 | 10.9 | 43.8 | 58.4 |
| SMINA | 1.4 | 6.2 | 15.2 | 10.3 | 30.1 | 46.7 | 0.8 | 2.6 | 12.7 | 8.5 | 45.3 | 63.5 |
| GLIDE | **0.5** | 8.3 | 29.5 | 15.7 | **43.4** | 45.7 | **0.3** | 4.9 | 28.5 | 14.8 | 45.4 | 50.4 |
| VINA | 4.5 | 9.7 | 19.9 | 13.4 | 13.2 | 26.7 | 1.7 | 5.5 | 18.7 | 11.2 | 29.8 | 47.9 |
| EQUIBIND-R | 2.0 | 5.1 | 9.8 | 7.4 | 25.1 | 49.0 | 1.4 | 2.6 | 7.3 | 5.8 | 40.8 | 66.9 |
| TankBind | 1.4 | **3.3** | **6.8** | **6.9** | 37.5 | **64.5** | 0.8 | **1.8** | **4.2** | **5.4** | **55.9** | **78.5** |

We also show the cumulative distribution plot for ligand RMSD and centroid distance in figure 5.

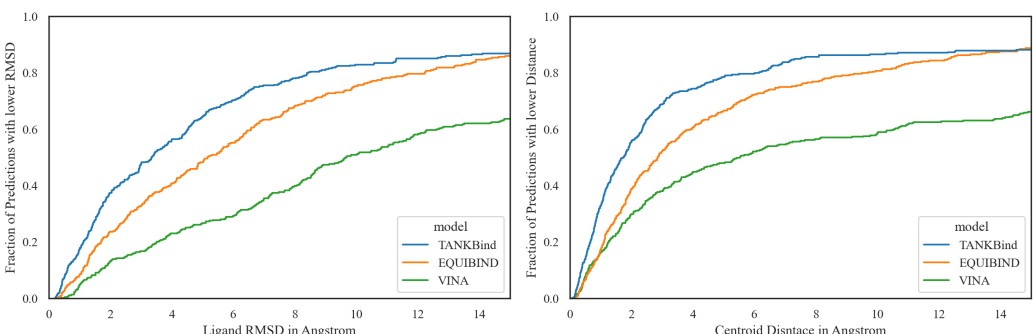

Figure 5: Blind re-docking. Estimator of the Cumulative Distribution Function (ECDF) plot for ligand RMSD (left) and Centroid Distance (right) based on the test set result. TankBind has a higher fraction of prediction with lower RMSD. The x axis of the figure stops at 15Å because comparison for larger RMSD is less meaningful, a RMSD of 15Å is not necessarily better than RMSD of 50Å.

## A.2   Blind re-docking new receptors

We also benchmark on the new receptors test set in which the proteins have not been seen in the training set, shown in table 6 and figure 6.

Table 6: Blind re-docking new receptors. All three models evaluated on 142 PDBs which is a subset of the original 363 PDBs. The proteins in this subset have not been seen in the training set.

| | **Ligand RMSD** | | | | | | **Centroid Distance** | | | | | |
| | Percentiles ↓ | | | | % Below Threshold ↑ | | Percentiles ↓ | | | | % Below Threshold ↑ | |
| **Methods** | 25% | 50% | 75% | Mean | 2Å | 5Å | 25% | 50% | 75% | Mean | 2Å | 5Å |
|---|---|---|---|---|---|---|---|---|---|---|---|---|
| VINA | 6.6 | 12.3 | 25.9 | 16.1 | 8.5 | 19.7 | 2.4 | 7.3 | 25.2 | 14.0 | 23.2 | 39.4 |
| EQUIBIND | 4.9 | 9.6 | 15.8 | 11.3 | 8.5 | 25.4 | 2.9 | 6.6 | 14.3 | 9.3 | 16.2 | 41.6 |
| TankBind | **2.6** | **4.6** | **9.2** | **8.8** | **20.4** | **53.5** | **1.4** | **2.5** | **6.0** | **6.9** | **37.3** | **73.2** |

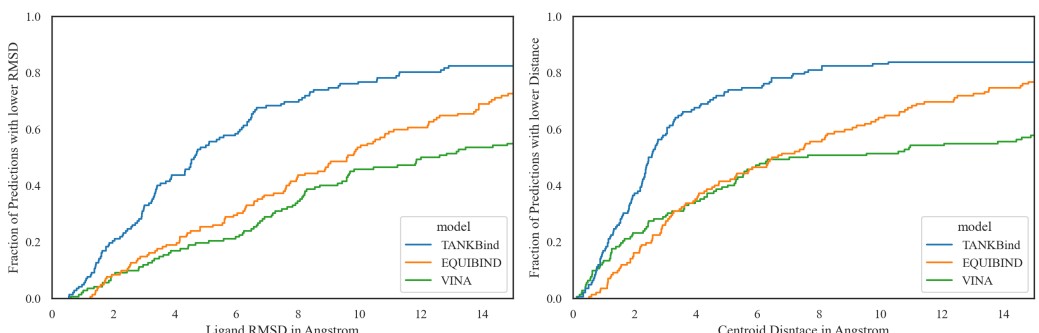

Figure 6: Blind re-docking new receptors. Estimator of the Cumulative Distribution Function (ECDF) plot for ligand RMSD (left) and Centroid Distance (right) from result evaluated on new receptors subset and with known ligand conformations. TankBind achieves a significantly better performance than the other two methods.

## A.3   Pseudo code for the trigonometry module and an illustrative figure for trigonometry updating.

The pseudo code for the whole trigonometry module is listed below. The whole module contains three sub-module: *TrigonometryUpdating*, *TriangleSelfAttention*, and *Transition*.

---

**Algorithm 1:** The `TrigonometryModule` function for updating $\mathbf{z}_{ij}$.

---

**Input:** $\{\mathbf{z}_{ij}\}, \{\mathbf{p}_{ik}\}, \{\mathbf{c}_{k'j}\}$

`// ` $\mathbf{z}_{ij} \in \mathrm{R}^{n \times m \times s}, \mathbf{p}_{ik} \in \mathrm{R}^{n \times n \times s}, \mathbf{c}_{k'j} \in \mathrm{R}^{m \times m \times s}$

**Output:** updated $\{\mathbf{z}_{ij}\}$

`// ` $\mathbf{z}_{ij} \in \mathrm{R}^{n \times m \times s}$

`/* Indices ` $i, k$ ` always operate on the residue dimension, ` $j, k'$ ` always on the compound dimension. ` $n$ ` is the number of protein nodes and ` $m$ ` is the number of compound nodes.                                          */`

1 **for** $\ell \leftarrow 0$ ***to*** $L$ **do**

2      $\{\mathbf{z}_{ij}\} \leftarrow \{\mathbf{z}_{ij}\} + \mathrm{Dropout}(\mathrm{TrigonometryUpdating}(\{\mathbf{z}_{ij}\}, \{\mathbf{p}_{ik}\}, \{\mathbf{c}_{k'j}\}))$

3      $\{\mathbf{z}_{ij}\} \leftarrow \{\mathbf{z}_{ij}\} + \mathrm{Dropout}(\mathrm{TriangleSelfAttention}(\{\mathbf{z}_{ij}\}))$

4      $\{\mathbf{z}_{ij}\} \leftarrow \mathrm{Tranistion}(\{\mathbf{z}_{ij}\})$

5 **end**

---

---

**Algorithm 2:** The `TrigonometryUpdating` function.

---

**Input:** $\{\mathbf{z}_{ij}\}, \{\mathbf{p}_{ik}\}, \{\mathbf{c}_{k'j}\}$

`// ` $\mathbf{z}_{ij} \in \mathrm{R}^{n \times m \times s}, \mathbf{p}_{ik} \in \mathrm{R}^{n \times n \times s}, \mathbf{c}_{k'j} \in \mathrm{R}^{m \times m \times s}$

**Output:** updated $\{\mathbf{z}_{ij}\}$

`// ` $\mathbf{z}_{ij} \in \mathrm{R}^{n \times m \times s}$

`/* Indices ` $i, k$ ` always operate on the residue dimension, ` $j, k'$ ` always on the compound dimension. ` $n$ ` is the number of protein nodes and ` $m$ ` is the number of compound nodes.                                          */`

1 $\mathbf{z}_{ij} \leftarrow \mathrm{LayerNorm}(\mathbf{z}_{ij})$

2 $\mathbf{p}_{ik} \leftarrow \mathrm{LayerNorm}(\mathbf{p}_{ik})$

3 $\mathbf{c}_{k'j} \leftarrow \mathrm{LayerNorm}(\mathbf{c}_{k'j})$

4 $\mathbf{p}_{ik} \leftarrow \mathrm{sigmoid}(\mathrm{Linear}(\mathbf{p}_{ik})) \odot \mathrm{Linear}(\mathbf{p}_{ik})$

5 $\mathbf{c}_{k'j} \leftarrow \mathrm{sigmoid}(\mathrm{Linear}(\mathbf{c}_{k'j})) \odot \mathrm{Linear}(\mathbf{c}_{k'j})$

6 $\mathbf{t}_{ij}, \mathbf{t}'_{ij} = \mathrm{sigmoid}(\mathrm{Linear}(\mathbf{z}_{ij})) \odot \mathrm{Linear}(\mathbf{z}_{ij})$

7 $\mathbf{o}_{ij} = \sum_{k=1}^{n} \mathbf{p}_{ik}\mathbf{t}_{kj} + \sum_{k'=1}^{m} \mathbf{t}'_{ik'}\mathbf{c}_{k'j}$

8 $\mathbf{g}_{ij} = \mathrm{sigmoid}(\mathrm{Linear}(\mathbf{z}_{ij}))$

9 $\mathbf{z}_{ij} \leftarrow \mathrm{Linear}(\mathrm{LayerNorm}(\mathbf{o}_{ij})) \odot \mathbf{g}_{ij}$

---

**Algorithm 3:** The `TriangleSelfAttention` function.

**Input:** $\{\mathbf{z}_{ij}\}$ // $\mathbf{z}_{ij} \in \mathbb{R}^{n \times m \times s}$

**Output:** updated $\{\mathbf{z}_{ij}\}$ // $\mathbf{z}_{ij} \in \mathbb{R}^{n \times m \times s}$

/* Indices $i, k$ always operate on the residue dimension, $j, k'$ always on the
    compound dimension. $n$ is the number of protein nodes and $m$ is the number of
    compound nodes. */

1   $\mathbf{z}_{ij} \leftarrow \text{LayerNorm}(\mathbf{z}_{ij})$

2   $\mathbf{q}_{ij}^h, \mathbf{k}_{ij}^h, \mathbf{v}_{ij}^h = \text{LinearNoBias}(\mathbf{z}_{ij})$ // $h \in \{1, ..., N_{head}\}$

3   $\mathbf{g}_{ij}^h = \text{sigmoid}(\text{Linear}(\mathbf{z}_{ij}))$

4   $w_{ijk'}^h = \text{softmax}_{k'}(\mathbf{q}_{ij}^{h\top} \mathbf{k}_{ik'}^h)$

5   $\mathbf{o}_{ij}^h = \mathbf{g}_{ij}^h \odot \sum_{k'=1}^{m}(w_{ijk'}^h \mathbf{v}_{ik'}^h)$

6   $\mathbf{z}_{ij} \leftarrow \text{Linear}(\text{concat}_h(\mathbf{o}_{ij}^h))$

---

**Algorithm 4:** The `Tranistion` function.

**Input:** $\{\mathbf{z}_{ij}\}$ // $\mathbf{z}_{ij} \in \mathbb{R}^{n \times m \times s}$

**Output:** updated $\{\mathbf{z}_{ij}\}$ // $\mathbf{z}_{ij} \in \mathbb{R}^{n \times m \times s}$

1   $\mathbf{z}_{ij} \leftarrow \text{LayerNorm}(\mathbf{z}_{ij})$

2   $\mathbf{z}_{ij} \leftarrow \text{Linear}(\text{ReLU}(\text{Linear}(\mathbf{z}_{ij})))$

---

Figure 7 shows the embeddings used to update a single interaction embedding, $z_{ij}$ between a protein node $i$ and a ligand node $j$.

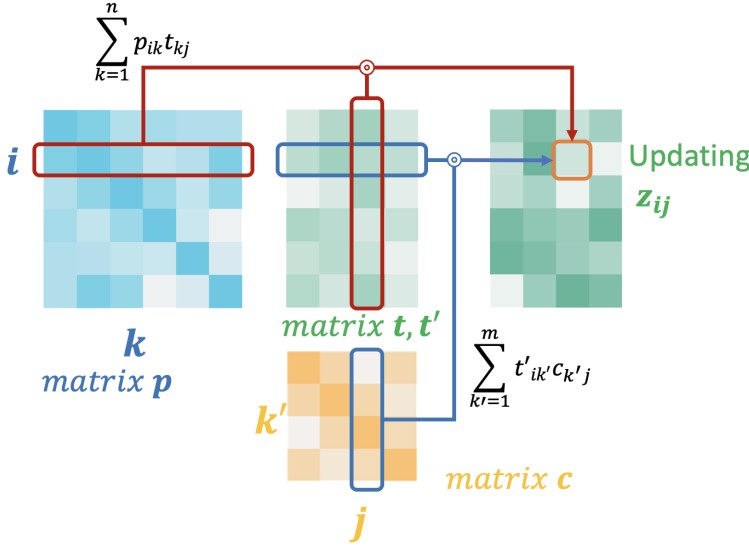

Figure 7: Illustrative figure for equation 1, $\tilde{\mathbf{z}}_{ij}^{(\ell)} = \mathbf{z}_{ij}^{(\ell)} + \Phi(\sum_{k=1}^{n} \mathbf{p}_{ik} \mathbf{t}_{kj}^{(\ell)} + \sum_{k'=1}^{m} \mathbf{t'}_{ik'}^{(\ell)} \mathbf{c}_{k'j}) \odot \mathbf{g}(\mathbf{z}_{ij}^{(\ell)})$

## A.4 Visualization of the gate functions and the self-attention map

To show the motivation behind the employment of the gate functions and self-attention, we have visualization them using an example, PDB 6HD6. Figure8 shows the output of the gate function at the last stacked layer for the native protein block with the compound FYH. Its resemblance to the inter-molecular distance map indicates the gate function is able to modulate the interaction based on the predicted distance.

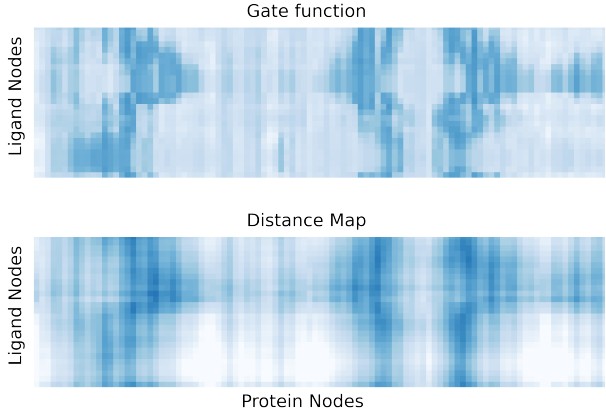

Figure 8: In upper figure, darker color means the gated value is closer to 1. lighter color means the gated value is closer to 0. In lower figure, darker color means the distance between the protein node and the ligand node is smaller, and lighter color means the distance is larger.

Figure9 shows the output of the self-attention map (3rd head) at the last stacked layer averaged over all protein nodes for the native protein block with compound FYH. The output resembles the compound intra-molecular distance map. This indicates that the self-attention module is aware of the compound conformation and is able to update the interaction embedding based on this.

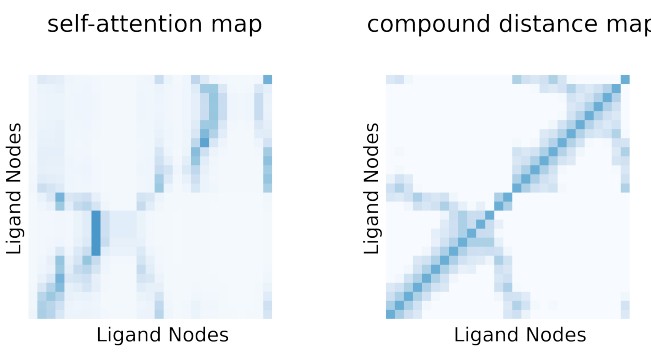

Figure 9: In upper figure, darker color means the self-attention value is larger. lighter color means the self-attention value is smaller. In lower figure, darker color means the distance between the ligand nodes is smaller, and lighter color means the distance is larger.

## B  Another example of TankBind finding the correct binding site when both the protein and the ligand are unseen.

A protein is unseen when the max protein sequence similarity (normalized Smith-Waterman alignment score) to the training set is less than 0.8. A compound is unseen when max compound similarity (Tanimoto Similarity of Morgan fingerprints) to the training set is less than 0.5. Figure 10 shows that, for PDB 6KQI, we correctly locate the true binding site on the protein, while the other two methods fail to do so under the same re-docking setting.

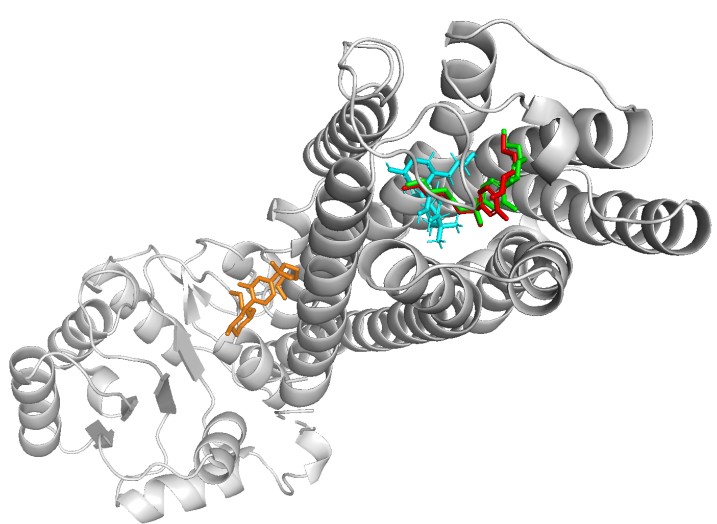

Figure 10: Visual inspection of PDB 6KQI. Another example of finding the native binding site when both the protein and the ligand are unseen. Crystallized ligand colored in green, TankBind prediction in red, EquiBind in cyan and Vina result in organ.

## C  Runtime details of different methods

The averaged runtime is shown in table 7. Baseline numbers are taken from EquiBind paper. The TankBind model has 1.8M parameters, comparable to EquiBind and GNINA, having 1.4M and 0.4M parameters respectively.

Table 7: averaged runtime per prediction for different methods.

| Methods | AVG. SEC. 16-CPU | AVG. SEC. GPU |
|---|---|---|
| QVINA-W | 49 | - |
| GNINA | 247 | 146 |
| SMINA | 146 | - |
| GLIDE(c.) | 1405* | - |
| VINA | 205 | - |
| EQUIBIND | 0.16 | 0.04 |
| TankBind | 0.54 | 0.28 |

## D Repeated runs of protein-ligand binding affinity prediction.

Table 8 provides more details than the figure in main text, including the mean and standard deviation of various methods across 3 experimental runs. Our model outperforms other models. For PIGNet and STAMP-DPI, we were unable to re-train the model, so we directly used the save-model provided by official repository for prediction.

Table 8: Comparison of predictive performance of ligand binding affinity using the PDBbind2020 dataset under time split.

| Methods | RMSE ↓ | Pearson ↑ | Spearman ↑ | MAE ↓ |
|---|---|---|---|---|
| **Sequence-based Methods** | | | | |
| TransformerCPI | $1.741 \pm 0.058$ | $0.576 \pm 0.022$ | $0.540 \pm 0.016$ | $1.404 \pm 0.040$ |
| MONN | $1.438 \pm 0.027$ | $0.624 \pm 0.037$ | $0.589 \pm 0.011$ | $1.143 \pm 0.052$ |
| **Complex-based Methods** | | | | |
| PIGNet※ | 2.640* | 0.511* | 0.489* | 2.110* |
| IGN | $1.433 \pm 0.028$ | $0.698 \pm 0.007$ | $0.641 \pm 0.014$ | $1.169 \pm 0.036$ |
| **Structure-based Methods** | | | | |
| HOLOPROT | $1.546 \pm 0.065$ | $0.602 \pm 0.006$ | $0.571 \pm 0.018$ | $1.208 \pm 0.038$ |
| STAMPDPI※ | 1.658* | 0.545* | 0.411* | 1.325* |
| TANKBind | $1.346 \pm 0.007$ | $0.726 \pm 0.007$ | $0.703 \pm 0.017$ | $1.070 \pm 0.019$ |

## E Additional ablation studies

Our ablation studies compose of mainly two categories. The first one is mainly associated with the framework of model, and the second one is associated with the training protocol. On the model side, TankBind-P is only doing the first summation over protein nodes inside the bracket of the equation 1, TankBind-C is only doing the second summation. We also included the results when the whole trigonometry module is only applied once, (single stack), and is completely removed, (no trig). For protein embedding, we tested using the pre-trained model TAPE to embed the protein instead of the GVP, and, for compound embedding, using GAT in place of GIN . On the side of training protocol, we tried replacing P2Rank binding sites with randomly selected binding sites, "TankBind-R", removing the random shift added to the center of protein block, "no random", and only training on the protein block contains the native ligand, "native only".

Table 9: Complete ablation results.

| | Ligand RMSD | | | | | | Centroid Distance | | | | | |
|---|---|---|---|---|---|---|---|---|---|---|---|---|
| | Percentiles ↓ | | | | % Below Threshold ↑ | | Percentiles ↓ | | | | % Below Threshold ↑ | |
| **Methods** | 25% | 50% | 75% | Mean | 2Å | 5Å | 25% | 50% | 75% | Mean | 2Å | 5Å |
| baseline | 2.45 | 3.96 | 7.67 | 7.43 | 19.28 | 61.71 | 0.87 | 1.74 | 4.22 | 5.51 | 56.47 | 77.41 |
| TankBind-R | 2.84 | 5.24 | 11.17 | 9.37 | 15.98 | 47.93 | 0.98 | 2.31 | 7.71 | 7.30 | 44.90 | 69.42 |
| native only | 3.01 | 7.14 | 21.49 | 12.92 | 17.08 | 41.05 | 1.05 | 4.68 | 20.23 | 11.34 | 37.47 | 51.52 |
| no shift | 2.65 | 3.94 | 7.73 | 7.57 | 19.56 | 58.68 | 0.79 | 1.75 | 4.53 | 5.60 | 55.10 | 76.58 |
| GIN to GAT | 2.48 | 4.05 | 7.72 | 8.27 | 19.01 | 57.02 | 0.82 | 1.66 | 4.19 | 6.23 | 56.47 | 78.51 |
| TAPE | 2.48 | 4.55 | 9.20 | 8.81 | 19.01 | 53.44 | 0.90 | 1.97 | 5.86 | 6.89 | 50.69 | 73.00 |
| TankBind-C | 2.38 | 4.47 | 8.36 | 8.20 | 19.56 | 54.82 | 0.93 | 1.87 | 5.41 | 6.27 | 53.17 | 73.28 |
| TankBind-P | 2.58 | 4.53 | 8.14 | 8.47 | 16.25 | 53.99 | 0.93 | 1.87 | 5.15 | 6.44 | 53.17 | 74.38 |
| single stack | 2.73 | 4.59 | 8.23 | 8.04 | 13.22 | 55.10 | 0.95 | 1.95 | 4.82 | 5.97 | 50.69 | 75.48 |
| no Trig | 3.34 | 5.26 | 8.56 | 8.73 | 4.13 | 47.93 | 1.25 | 2.22 | 5.01 | 6.44 | 44.08 | 74.93 |

## F    Example of the existence of equally valid binding sites that confuses the result

PDBbind curated the raw PDB file by only preserving a single ligand. In a few cases, when two or more identical chains are crystallized together, there are more than one valid binding site for the ligand. Here, we show an example with PDB 6MO9 in figure 11.

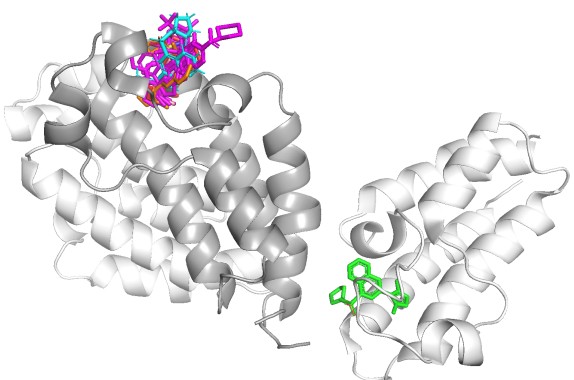

Figure 11: An example of equally valid binding sites that confuses the result. Green is the ligand preserved by PDBbind, but there is an equally valid binding site for each chain. PDB 6MO9.

## G    Details about protein segmentation

Protein graph is segmented for two main reasons: computational memory efficiency and biological functional generalization. One the computational side, since the size of proteins has a large variation, ranging from a few dozen amino acids to more than 3000 amino acids, the memory consumption to represent the protein and the interaction between protein and ligand could easily exceed the capacity of a typical GPU. On the biological side, large protein typically have multiple domains. Each protein domain, typically of size 200 amino acids, folds and interacts with ligand independently from the rest. Also, protein domains, as the building blocks of proteins, are more evolutionary conserved which means that a model explicitly learning on a domain level could generalize better to new domains. Each block is a sphere of radius 20Å typically includes about 200 amino acids, in line with the size of a protein domain. Block of radius 20Å is large enough to enclose the drug molecules, which is usually less than 15Å long, and small enough to be memory efficient.

We tried two ways of segmenting the whole protein. First approach is random segmentation; We randomly select an protein node, and use this node as the block center. But this approach is not efficient, since the binding site on protein has certain characteristic, more hydrophobic for instance. In second approach, we used a ligand-agnostic method, P2rank v2.3 (trained model was released on 2018) Krivák and Hoksza [2018] to identify possible ligand binding sites, and use the centers of those potential binding sites as the block centers. For some small proteins, no binding site is identified, we therefore add an extra protein block located at the center of the whole protein. During training, we also add an extra protein block centering at the centroid of the co-crystallized ligand.

Despite P2rank assigns a score to each predicted binding site on the protein, the scores are fixed regardless of the interacting ligand because of the ligand-agnostic nature of P2rank. If we simply use the center of most probable binding site predicted by P2rank as the center of interaction block, this interaction block encloses the ligand for 73% of the test set. Our method, being a ligand-dependent method, improve the rate to 90%.

# H  Systematically analysis of the PDBbind dataset

In order to examine whether our model has the capability to place the compound to a unseen binding site, we aligned all the training protein-ligand complex structure with the same protein, defined by having the same *UniProt ID*, to the test set protein. *Kalign* is used to align the protein sequences first, and *Superimposer* function within package *BioPython* is used to align the structures. After the alignment, we computed centroid distance between aligned compounds and centroid of the test set compound. We found three cases in total having the centroid of all aligned training set compounds at least 10Å away from the centroid of test set ligand. They are PDB 6HMY, 6MO9 and 6K1S. With a visual examination of these PDBs along with aligned training set PDBs, we found that the, for PDB 6HMY and 6MO9, the apparently unseen binding site are caused by the process of crystallization and multimeric nature of certain proteins. For PDB 6MO9, the single chain protein having two identical binding sites due to packing during crystallization. For PDB 6HMY, the protein is a pentamer which means there are actually 5 identical binding sites for the complete protein complex. But in PDB 6K1S, we found a genuine unseen site. We found three training set PDBs having the same protein: 4X60, 4X61, 4X63. In order to remove the possibility that a close homolog exists, we computed the normalized Smith-Waterman alignment score, and found no homolog with score above 0.8 for PDB 6K1S other than the three PDBs with identical protein mentioned beforehand. Compound similarity is computed based on the Tanimoto Similarity of their Morgan fingerprint using RDKit.

# I  Hyper-parameters

The embedding sizes of the embedding of protein blocks and compound embedding are 128. The channel sizes of distance embedding are also 128. The trigonometry module is stacked 5 times. Transition module is a multilayer perceptron, where the hidden channel size is four times the input channel size. Dropout rate is set to 25%. layernorm is applied after each transition.

# J  Training details

During training, data is augmented in two ways. First the protein blocks that do not bind to the specific compound are used as decoys. The constrastive loss function is designed to ensure the compound binds weaker to those decoys than the native protein block. The margin, $\epsilon$ in constrastive loss is set to 1, corresponding to 1 order of magnitude in binding concentration. A protein block encloses the ligand when it covers more than 90% of the native interaction. Second, the model will see a slightly different protein block for every training data because the center of block will have a random shift of -5Å to 5Å, drawn from the uniform distribution, in all three axes. In bind re-docking, the native conformation is given as input, while in bind self-docking setting, the local atomic structures (LAS) mask, as defined in section 3.5 is applied to the compound node pair-distance map. The compound node pair distance map is based on the native conformation during training and based on the conformation generated by RDKit during testing. Our model include the models in ablation studies are trained for 200 epochs, after which no performance gain was observed. The model with the lowest validation loss was chosen as the best model. Each epoch has 20,000 randomly sampled block-ligand pairs. The total training process takes about 50 hours on a single NVIDIA RTX 3090 GPU. We use Adam optimizer with a constant learning rate of 0.0001.

# K  Implementation details of baselines for drug-protein binding structure prediction

**Vina**  AutoDock Vina v1.2.3 is downloaded from `https://github.com/ccsb-scripps/AutoDock-Vina`. We follow the tutorial listed in `https://autodock-vina.readthedocs.io/en/latest/docking_basic.html` and use the center of ligand as the box center. The box size is set to 100Å and exhuastiveness is set to 32.

**EquiBind**  EquiBind is downloaded from `https://github.com/HannesStark/EquiBind`. We follow the instruction and use saved model as listed in the GitHub. Our result slightly differs from the reported value. It could be due to a version change made by the developer, since its still an

active repository. In the setting of "new receptors", following the same procedure, We got 142 "new receptors" PDBs while EquiBind got 144 (exact list not provided). We estimated that the 142 version and 144 version will affect the reported value by less than 2%. The result for other baselines are copied directly from the EquiBind paper for ease of comparison.

## L    Implementation details of baselines for binding affinity prediction

**TransformerCPI**    We downloaded the code from the official repository `https://github.com/lifanchen-simm/transformerCPI`. we changed from the default classification task to regression task and switched to the PDBbind2020 dataset with time split, word2vec model is also retrained to extract sequence features based on the new dataset.

**MONN**    We downloaded the code from the official repository `https://github.com/lishuya17/MONN`. The authors did not use a separate validation set, but instead used a clustering-based cross-validation strategy. We switched the data split mode to time split and repeated the original authors' data preprocessing steps on PDBbind2020 dataset.

**PIGNet**    We downloaded the code and best save-model (best performance on CASF2016 benchmark) from their official repository `https://github.com/ACE-KAIST/PIGNet`. Due to the lacking of pre-processing scripts for data augmentation, we were unable to re-train the model using PDBbind2020. Instead, we used the best save-model presented by the authors. The result could be improved with additional data augmentation on the whole dataset instead of the PDBbind2019 refined set currently used for training.

**IGN**    We downloaded the code from their official repository `https://github.com/zjujdj/InteractionGraphNet/tree/master`. The authors used PDBbind V2016 as an experimental dataset. We switched the data split mode to time split and repeated the data pre-processing protocol used by the authors on PDBBind2020.

**HOLOPROT**    We downloaded the code from their official repository `https://github.com/vsomnath/holoprot`. The authors used PDBbind2019 refined set as an experimental dataset split by ligand scaffold and protein sequence. We followed the original authors' data pre-processing on PDBbind2020 and calculated the multi-scale representation of proteins. The model was retrained on this new dataset with the default setting.

**STAMP-DPI**    We downloaded the code from their official repository `https://github.com/biomed-AI/STAMP-DPI`. The authors used PDBbind2016 general set as training set. We followed the original data pre-processing and performed on the split PDBbind 2020 test set. We were unable to extract all features required for training due to time constraints. The released model was employed to evaluate on the test set without re-training.

## M    Potential negative societal impacts

The greatest possible risk comes from virtual screening, such as practitioners blindly trusting the affinity predictions of learned models. Inspection of the predicted protein-ligand structure could help alleviate this issue, and fundamentally, further model Improvements and data augmentation to binding affinity prediction are essential to address this.