# OpenReview forum: "TANKBind: Trigonometry-Aware Neural NetworKs for Drug-Protein Binding Structure Prediction"
_NeurIPS.cc/2022/Conference — NeurIPS 2022 Accept_

### Official Review · Reviewer_DuMw · 2022-06-27

**Rating:** 5
**Confidence:** 5
**Soundness:** 3 good
**Presentation:** 3 good
**Contribution:** 2 fair

**Summary:**

In this manuscript, the authors propose Trigonometry-Aware Neural networKs for binding structure prediction, TANKBind, that builds trigonometry constraint as a vigorous inductive bias into  the model and explicitly attends to all possible binding sites for each protein by  segmenting the whole protein into functional blocks.  The algorithm is trained by minimizing a novel contrastive loss with local region negative sampling. The model is able to predict both the binding poses and binding affinity between protein targets and small molecules.

**Questions:**

Questions:
1. What is the overlap between the training data of P2rank and Tankbind? It is necessary to check whether there is information leak for training and test datasets.
2. I did not find the definition of ‘blind’
3. It might take biologists decades to verify a new drug target. A more practical experiment is to show the generalizability to the new compounds instead of new protein targets.
4. It seems that the predicted protien-ligand complex is not feasible even using the trigonometry update function. That is why the authors need to use Eq 5 and 6. Please add discussion that why this component is still necessary when trigonometry update function is applied.

**Ethics Review Area:**

["I don’t know"]

**Limitations:**

No obvious limitations and potential negative societal impact

**Strengths And Weaknesses:**

Strengths: So far as I know, this is the first algorithm which adopts the geometric consistency to the binding of proteins and small molecules. Another contribution is the loss function which treats the non-binding sites on the same protein as the negative training data.

Weakness: First, I want to emphasize that geometric consistency (trigonometry) is first proposed by AlphaFold2. The embedding z_ij for a pair of amino acids i,j is updated by another two embeddings z_ik and z_kj to satisfy the triangle constraints. Tankbind changes the transformer of AlphaFold2 to another gating function. In theory, it is easy to modify the StructureModule of AlphaFold2 to also fit this task. For me, this decreases the novelty of this work. Second, my major concern is that it is not clear what is the impact of P2rank which predicts the functional site of proteins. VINA-based methods and EquiBind both identify the binding sites by themselves. If AUTODOCK predicts a wrong binding site, it will yield a super low score. However, what if the authors first run the P2rank to predict the binding sites and then feed the predicted binding sites to AUTODOCK and EquiBind? The ablation study also shows that P2rank is the most important component and removing the P2rank will decrease the performance.

---

> ### Author Response · Authors · 2022-08-02
> **Response to Reviewer DuMw (part 1/2)**
>
> We appreciate the reviewer's careful review of our paper, positive feedback and constructive comments. We thank the reviewer for acknowledging us to be the first in adopting the geometric consistency to the binding of protein and small molecules. We have addressed the reviewer's specific concerns and questions below.
>
>
> > Geometric consistency (trigonometry) is first proposed by AlphaFold2. it is easy to modify the StructureModule of AlphaFold2 to also fit this task.
>
> The trigonometry module (part of the Evoformer module, not the StructureModule) of AlphaFold2 (as illustrated in their Supplementary Figure 6) is designed to update a N by N square matrix. A simple extension of the module to a N by M matrix, as in the case of protein-ligand pair representation, will miss the valuable information stored inside the intra-molecular pair representations of ligand and protein themselves. Our trigonometry module, as described in Eq. (1), is based on the similar idea of AlphaFold2, but is a novel extension of the idea of geometric consistency. Our trigonometry module updates the pair representation not only using itself but also using the protein matrix p and the compound matrix c with extra adaptive gate functions. We have included a schematic illustration of our trigonometry module (Figure 7) in appendix.
>
>
> > my major concern is that it is not clear what is the impact of P2rank which predicts the functional site of proteins. what if the authors first run the P2rank to predict the binding sites and then feed the predicted binding sites to AUTODOCK and EquiBind? The ablation study also shows that P2rank is the most important component and removing the P2rank will decrease the performance.
>
> P2Rank is a ligand-agnostic method. It provides a list of potential binding sites and corresponding scores. But these scores are constant for all ligands because they are computed entirely based on the protein properties. We use the top-scored pocket predicted by P2Rank and run AUTODOCK as suggested by the reviewer. Result is shown in table below, the mean ligand RMSD is 12.54Å which is slightly better than the original 14.7Å but still much worse than TankBind's 7.4Å. Same approach cannot be applied for EquiBind as it was trained (and made predictions) on the whole proteins and did not support prediction with a certain bounding box (https://github.com/HannesStark/EquiBind/issues/22).
> Regarding to the ablation study where blocks are extracted using randomly picked centers instead of the P2rank predicted centers, the deterioration of performance is mainly because, in many cases, none of the ten randomly picked blocks contains the native binding site, so the model is unable to locate the ligand to the right place. Briefly, P2rank could help us to focus on those functional blocks that have a chance to be the true binding site, but it is not a good method in selecting the exact binding site for a specific query compound.
>
>
> Method | Mean Ligand RMSD | \% below threshold 2Å  | \% below threshold 5Å  | Mean Centroid Distance
> -------|--------|------- | ------- | -------
> VINA(P2rank pocket)  | 12.54  | 3.64 | 24.57 | 9.58
>
> > What is the overlap between the training data of P2rank and Tankbind? It is necessary to check whether there is information leak for training and test datasets.
>
> The P2rank paper was published in 2018 and the model was trained on CHEN11 dataset(an extremely small dataset with 251 complex structures chosen from the protein data bank). None of these structures is in our test set (structures deposited to protein data bank after 2019). Thus, there is no information leak for the test set.
> We have modified the original sentence to "x_o is the center of the functional block predicted by a widely-used ligand-agnostic method, P2rank (published in 2018)" in the main text to clarify this.
>
> > I did not find the definition of ‘blind’
>
> 'Blind' means no information regarding to the complex structure is given during prediction, in contrast to the setting where the binding site is given.
> We added a line "In blind docking, the protein binding site is assumed unknown" to the main text.

---

> > ### Author Response · Authors · 2022-08-02
> > **Response to Reviewer DuMw (part 2/2)**
> >
> >
> > > It might take biologists decades to verify a new drug target. A more practical experiment is to show the generalizability to the new compounds instead of new protein targets.
> >
> > Interesting points. In fact, the majority of the 363 ligands (87\%) of the test set are new compounds. 152 (42.4\%) have the max compound fingerprint Tanimoto similarity against training set less than 0.5, which can be deemed as novel scaffolds. We computed the statistic for those 152 cases. The result (shown below) is about the same as the result of the complete test set (363).
> >
> > Method | Mean Ligand RMSD | \% below threshold 2Å  | \% below threshold 5Å
> > -------|--------|------- | -------
> > TankBind | 7.43 | 19.28 | 61.71
> > EquiBind | 8.13 | 3.31 | 39.12
> > VINA   | 14.65  | 5.51 | 21.21
> >
> >
> >
> > > It seems that the predicted protien-ligand complex is not feasible even using the trigonometry update function. That is why the authors need to use Eq 5 and 6. Please add discussion that why this component is still necessary when trigonometry update function is applied.
> >
> > Trigonometry update function is used to create a realistic interaction matrix and cannot output coordinates directly (also true for AlphaFold2). From this interaction matrix to ligand coordinates, we adopt an optimization approach (Eq 5 and 6) to find ligand coordinates that satisfy the most constraints based on the predicted interaction matrix and local atomic structure, like the first generation AlphaFold. We could also use a structure module similar to AlphaFold2, but our preliminary attempt to directly output coordinates fails to ensure the directly outputted ligand having realistic conformation, probably due to limited training data. We will leave this as future work for now.

---

> > ### Comment · Reviewer_DuMw · 2022-08-07
> > **Re: Response to Reviewer DuMw**
> >
> > 1. To clarify the contribution related to geometric consistency, AlphaFold2 achieves the geometric consistency on a single protein and TankBind use the same technique on protein-ligand pair. Is my understanding correct? If I adapt the same technique (geometric consistency) to the prediction tasks of protein-RNA complex, protein-DNA complex, RNA 3D structure, RNA-ligand complex, do the authors agree they have contributions on the machine learning model construction and should be accepted as 4 different NeurIPS papers (of course I will also adapt other established software such as P2rank in the pipeline)?
> >
> > 2. Second, my question about the P2rank is fully addressed. Thanks.

---

> > > ### Author Response · Authors · 2022-08-08
> > > **geometric consistency**
> > >
> > > Thank you for your constructive feedback. Regarding technical contribution related to geometric consistency, we would like to make the following clarifications.
> > >
> > > "Geometric consistency", in our opinion, is more of a general idea than a plug-and-play module. To apply it to the task of interest, we need to implement a machine learning model that suits the task.
> > > Our work is the first implementation of "geometric consistency" for the modeling of the inter-molecular interaction in a heterogenous system and demonstrates its usefulness by achieving significant performance gain over baselines.
> > > We are glad like that our success in setting the new state-of-the-art for the protein-ligand problem suggests the possibility of performance gain in other tasks as mentioned by the reviewer.
> > >
> > > The tasks of protein-DNA/RNA and RNA structure prediction are all fundamental challenges facing science, and probably also requires ingenious designs and tailor-made implementation of "geometric consistency". For example, in RNA modeling, we need to decide whether one node represents a nucleobase or a single atom, or other choices. In addition, special consideration is required to learn from even less data available for the problem of protein-DNA/RNA prediction. If those work also achieve above SOTA performance, like TankBind does, it will be a sign that more exploration around the implementation of "geometric consistency" will be rewarding, and could be an interesting research area for modeling of macromolecule system. Such work, we believe, deserve to be published in a high impact conference like NeurIPS.

---

### Official Review · Reviewer_DdyQ · 2022-07-11

**Rating:** 5
**Confidence:** 3
**Soundness:** 2 fair
**Presentation:** 3 good
**Contribution:** 2 fair

**Summary:**

Based on Geometric Vector Perceptron (GVP) for protein modelling and  Graph Isomorphism Network (GIN) for drug processing, the paper proposes a method for drug-protein binding structure prediction.  The main idea is to build a trigonometry constraint and explicitly attend to all possible binding sites for each protein by segmenting the whole protein into functional blocks.

**Questions:**

1. Although I recognize that the paper is mostly technically sound, such as using the sum of local protein and protein information to reflect the interaction, I feel some details are not well motivated. Specifically, according to Eq. (1) and Eq. (2), the method employs many gates, whose motivation is not very clear. Moreover, it could be better to visualize the gates to show some insights.


2. The framework includes a  self-attention module to modulate the interaction between a protein node and all compound nodes by taking the whole interaction between this protein node and all compound nodes into consideration.  As a  self-attention module, it could be better to visualize the self-attention map via a few examples to verify the motivation.


3.  Trigonometry is the main contribution or novelty and is similar to or inspired by the Evoformer module used in AlphaFold2. The difference should be highlighted.

**Ethics Review Area:**

["I don’t know"]

**Limitations:**

NA.

**Strengths And Weaknesses:**

1. The paper is well written. Readers can learn a lot about drug-protein interaction from the paper.

2. The method is mostly technically sound.

3. Comprehensive experiments show the effectiveness of the proposed method.

---

> ### Author Response · Authors · 2022-08-02
> **] Response to Reviewer DdyQ**
>
>
> We thank the reviewer for the time taken to review our work and constructive feedback provided. We are glad that the reviewer find our paper "well written" and "technically sound". We have addressed the reviewer's specific questions below. The updated paper has been uploaded to Open Review.
>
>
> > Although I recognize that the paper is mostly technically sound, such as using the sum of local protein and protein information to reflect the interaction, I feel some details are not well motivated. Specifically, according to Eq. (1) and Eq. (2), the method employs many gates, whose motivation is not very clear. Moreover, it could be better to visualize the gates to show some insights.
>
> Physical interactions between ligands and proteins are relatively short-ranged because the relatively long-ranged electrostatic force is shielded by salt ions in solution.[1] Therefore, those gate functions are employed to decrease the influence of protein-ligand nodes pair when the inferred distance between them is large. We have added a section (Appendix A.4) that visualize the output of a gate function using an example (PDB 6HD6). The visualized output of the gate function indeed looks like the native inter-molecular distance matrix.
>
> > The framework includes a self-attention module to modulate the interaction between a protein node and all compound nodes by taking the whole interaction between this protein node and all compound nodes into consideration. As a self-attention module, it could be better to visualize the self-attention map via a few examples to verify the motivation.
>
> Good points. In the added section (Appendix A.4), we also include a visualized self-attention map for the same example. The map looks like the compound intra-molecular distance map. It is a reassuring sign because, in our setting, the ligand conformation is unknown, and the self-attention could predict and use the information about ligand conformation to update the protein-ligand interaction as designed to.
>
> > Trigonometry is the main contribution or novelty and is similar to or inspired by the Evoformer module used in AlphaFold2. The difference should be highlighted.
>
> We thank the reviewer for this great question.
>
> a) First, the trigonometry module in our paper is, to the best of our knowledge, the first inter-molecular modeling method that captures the geometry consistency between two heterogeneous systems. This is a novel extension of the intra-molecular trigonometry module in AlaphaFold2 (Evoformer). Traditional solutions[2,3,4] to the problem of drug-protein binding structure prediction focus on designing sophisticated two-body terms to approximate the protein-compound interaction which is, however, an intrinsically many-body interaction. Our trigonometry-aware neural network is the first neural network-based method that is capable of explicitly learning the many-body effects while maintaining a low computational cost for this important research area of protein-compound interaction.
>
> b) Second, a simple extension of the Evoformer cannot handle the heterogeneous systems. The pair representation updated by the trigonometry module of AlphaFold2 is a square matrix. To predict the complex structure of protein-ligand pair, our pair representation is not a square matrix but a N by M matrix. A direct application of trigonometry module of AlphaFold2 will miss the valuable information stored inside the intra-molecular pair representations of ligand and protein themselves. Therefore, we updated the the pair representation not only using itself but also using the protein matrix p and the compound matrix c with extra adaptive gate functions (Eq. (1), also a schematic illustration in figure 7).
> In addition, the new trigonometry module is part of our novel design to tackle the drug binding problem. As pointed out by Reviewer YkVW, our model achieves the state-of-the-art performance in drug-protein binding structure prediction by an innovative combination of the trigonometry module, a divide-and-conquer strategy for dealing with proteins, and the joint training of binding pose and affinity with contrastive losses.
>
>
> [1] Erbaş, Aykut, Monica Olvera De La Cruz, and John F. Marko. "Effects of electrostatic interactions on ligand dissociation kinetics." Physical Review E 97.2 (2018): 022405.
>
> [2] Trott, Oleg, and Arthur J. Olson. "AutoDock Vina: improving the speed and accuracy of docking with a new scoring function, efficient optimization, and multithreading." Journal of computational chemistry 31.2 (2010): 455-461.
>
> [3] Friesner, Richard A., et al. "Glide: a new approach for rapid, accurate docking and scoring. 1. Method and assessment of docking accuracy." Journal of medicinal chemistry 47.7 (2004): 1739-1749.
>
> [4] Méndez-Lucio, Oscar, et al. "A geometric deep learning approach to predict binding conformations of bioactive molecules." Nature Machine Intelligence 3.12 (2021): 1033-1039.

---

> > ### Comment · Reviewer_DdyQ · 2022-08-08
> > **Comments on Authors' Rebutal**
> >
> > Thank the authors for the detailed responses and the revised supplementary. My concerns are addressed.

---

> > > ### Author Response · Authors · 2022-08-08
> > > **Thanks for your constructive comments!**
> > >
> > > Thanks for your constructive comments and encouragement. We are pleased that our responses have addressed your concerns. Please let us know if there is anything that could help improve your rating. Thank you!

---

### Official Review · Reviewer_YkVW · 2022-07-11

**Rating:** 7
**Confidence:** 3
**Soundness:** 4 excellent
**Presentation:** 3 good
**Contribution:** 4 excellent

**Summary:**

This paper proposes a Trigonometry-Aware Neural Network (TankBind) for drug-protein binding site prediction. In the proposed method,  the protein is first segmented into several functional blocks by an out-of-shelf model. Then, a trigonometry module inspired by Evoformer of AlphaFold2 is applied to update the pairwise embedding between protein and ligand. The contrastive loss with local region negative sampling is applied to jointly optimize the binding position and affinity. The model could achieve state-of-the-art performance in both binding position prediction and binding affinity prediction tasks.

**Questions:**

- Is the ligand conformer fixed after the 3D initialization?
- How is the margin value epsilon determined in Equation 4?
- In the binding affinity prediction experiment, is the same loss function (Eq. 4) applied? Is the binding site given as known information? If so, there is no need to apply the contrastive loss.
- In the docking experiments, are methods listed in Table 2 all deterministic? If not, have you run these methods multiple times to make results more reliable?

**Limitations:**

The authors discuss several future work directions in the conclusion section. There is not negative social impact.

**Strengths And Weaknesses:**

Strengths:
- The proposed method is novel, especially combining different designs together: breaking proteins into functional blocks, applying trigonometry module to update distance maps, performing affinity and binding poses joint training with contrastive losses.
- The paper is well-written and easy to follow.
- The proposed method shows superior empirical performance over existing baselines.

Generally, I think it's a very good paper and didn't see any major weakness. Only some minor questions left,  see details in Questions.

---

> ### Author Response · Authors · 2022-08-02
> **Response to Reviewer YkVW**
>
>
> We thank the reviewer for the time taken to review our work and constructive feedback provided. We are glad that the reviewer enjoys the work, believes that the work is "novel" and the presentation is “well-written and easy to follow”. We have addressed the reviewer's specific questions below. The updated paper has been uploaded to Open Review.
>
> > Is the ligand conformer fixed after the 3D initialization?
>
> In the re-docking setting (as reported in Appendix A), the ligand conformation is fixed while in the self-docking setting (as reported in the main text), the ligand conformation is not fixed.
> We have modified the sentence "We start with a real-world blind self-docking experiment." in the main text to "We start with a real-world blind self-docking experiment, in which the ligand conformation is not fixed, and the result of re-docking experiment, in which the native ligand conformation is given, is reported in Appendix A." to explicitly state the difference.
>
>
> > How is the margin value epsilon determined in Equation 4?
>
> The margin value epsilon is chosen such that the ligand binds at least one order of magnitude, as measured in unit of concentration, stronger to the native block than to non-native blocks. One order of magnitude is chosen mainly for the following two reasons.
> First, we do not know the exact binding affinity to non-native blocks, other than it is weaker than the affinity to the native block, a larger epsilon could impose constraints too strong to be realistic. Second, when the epsilon is too small, the native block could not be distinctly differentiated from non-native blocks. We have experimented with epsilon 2 and 0.5, both produced results slightly worse than the default value of 1.
>
>
> > In the binding affinity prediction experiment, is the same loss function (Eq. 4) applied? Is the binding site given as known information? If so, there is no need to apply the contrastive loss.
>
> Yes, the same loss is applied. The binding affinity and position are optimized jointly. The native binding site is not given as known information in all of our test experiments, as the binding site for a specific molecule is not available in many realistic situation [1].
>
>
> > In the docking experiments, are methods listed in Table 2 all deterministic? If not, have you run these methods multiple times to make results more reliable?
>
>
> They are not deterministic, but we have run VINA, EquiBind and TankBind three times and the mean ligand RMSD is reported below. The standard deviation of the mean ligand RMSD is smaller than the difference between these methods.
>
> Method | Repeat 1 | Repeat 2  | Repeat3
> -------|--------|------- | -------
> TankBind | 7.43 | 7.5 | 7.41
> EquiBind | 8.2 | 8.13 |  8.13
> VINA   | 14.65  | 13.3 | 13.2
>
>
> [1] Song, Minsoo, and Gil Tae Hwang. "DNA-encoded library screening as core platform technology in drug discovery: its synthetic method development and applications in DEL synthesis." Journal of medicinal chemistry 63.13 (2020): 6578-6599.

---

> > ### Comment · Reviewer_YkVW · 2022-08-07
> > **Thank you for the response**
> >
> > Thank you for the response. It well addressed my concerns.

---

### Meta-Review · Area_Chair_n42V · 2022-08-29

**Recommendation:** Accept
**Confidence:** Less certain

**Metareview:**

This is a bordeline paper.

All three reviewers liked the paper and appreciated the feedback from the authors. The numerical score are 7, 5 and 5. The two 5s appear a bit on the low side given the comments.

The paper tackles the difficult problem of protein-ligand docking with a geometrical GNN approach inspired in part by AlphaFold. This is an important problem and the paper presents a novel approach in a clear and convincing way. Acceptance is therefore recommended.

**Award:**

No

---

### Decision · Program_Chairs · 2022-09-14

Accept